# Super-resolution imaging of light–matter interactions near single semiconductor nanowires

Eric Johlin[1], Jacopo Solari[1], Sander A. Mann[1], Jia Wang[1], Thomas S. Shimizu[1] & Erik C. Garnett[1]

Nanophotonics is becoming invaluable for an expanding range of applications, from controlling the spontaneous emission rate and the directionality of quantum emitters, to reducing material requirements of solar cells by an order of magnitude. These effects are highly dependent on the near field of the nanostructure, which constitutes the evanescent fields from propagating and resonant localized modes. Although the interactions between quantum emitters and nanophotonic structures are increasingly well understood theoretically, directly imaging these interactions experimentally remains challenging. Here we demonstrate a photoactivated localization microscopy-based technique to image emitter-nanostructure interactions. For a 75 nm diameter silicon nanowire, we directly observe a confluence of emission rate enhancement, directivity modification and guided mode excitation, with strong interaction at scales up to 13 times the nanowire diameter. Furthermore, through analytical modelling we distinguish the relative contribution of these effects, as well as their dependence on emitter orientation.

[1] FOM Institute AMOLF, Science Park 104, 1098 XG Amsterdam, The Netherlands. Correspondence and requests for materials should be addressed to E.J. (email: johlin@alum.mit.edu) or to E.C.G. (email: garnett@amolf.nl).

The near-field interactions of a quantum emitter with a nanostructure can be summarized largely by three pheonomena: first, the excitation of guided and resonant modes within the structure, second, the directing of emission through interference with scattering from the nanostructure and third, the modification of the total emission rate of the quantum emitter.

Nanowires have proven to be an ideal model system for nanophotonic exploration as they are both geometrically simple enough to be amenable to analytical analysis, while also expressing strong coupling to all three of these channels, as depicted by the inset in Fig. 1a. They support both localized resonances[1] and guided modes[2], which modify the local density of optical states (LDOS) at the location of the emitter (allowing for emission rate and efficiency enhancements), as well as its radiation pattern (allowing for improved directional emission)[3].

This ability to control both the emission and absorption of light has led to a huge number of applications throughout the fields of light-emitting diodes[4], nanoscale lasers[5], photovoltaic[6–8] and photodetection devices[9], single-photon sources[3,10] and enhanced biological imaging[11]. Despite this wide interest, imaging these phenomena within the near field of such structures has proven quite challenging.

Super-resolution fluorescence measurements provide the intriguing possibility of directly imaging near-field optical interactions while eliminating the need for either external probes[12–14] (which can perturb the local environment) or an electron-beam excitation[15–18] present in other methods. Previous measurements have generally examined plasmonic properties using dye functionalized onto the surface of metallic nanostructures[19–22], diffusing at low densities around a hotspot[23] or immobilized in a plasmonic lattice[24]. These studies, however, do not map the near-field interactions away from the surface of a single structure. Furthermore, these techniques have not been demonstrated for semiconductor structures, likely because the lower interaction strength and higher absorption necessitates large sampling ensembles, higher signal-to-noise ratios or both.

Here we demonstrate a super-resolution technique utilizing photo-activated localization microscopy (PALM) of fluorophore molecules in a liquid-phase medium to allow mapping of point-emitter-nanostruture interactions with an ∼25 nm resolution over large (hundreds of square microns) areas. While traditional PALM measurements determine solely the location of molecules on an unknown structure to probe the physical geometry of the sample[25–27], herein we use a solution of fluorophore dye, providing local measurements at all positions. This is similar to the point accumulation for imaging in nanoscale topography (PAINT) technique[28]. By measuring the modulation of the observable brightness as well as location, we interrogate the near-field optical interactions between quantum emitters and nanostructures, providing direct imaging of the strong, extended coupling between dipole-like sources and nanoscale antennas.

## Results

**Dipole-nanowire interactions.** We begin by computationally exploring the interaction of 648 nm wavelength emitters with a 75 nm diameter silicon nanowire, as shown in Fig. 1a–c. These interactions can be calculated using commercial software packages or, as in this case, solved analytically using a Green's function approach[29]. For this size and wavelength, absorption in the nanowire is minimal and can be neglected (Supplementary Note 1), such that we only have to consider the scattering and excitation of waveguide modes. First, Fig. 1a demonstrates

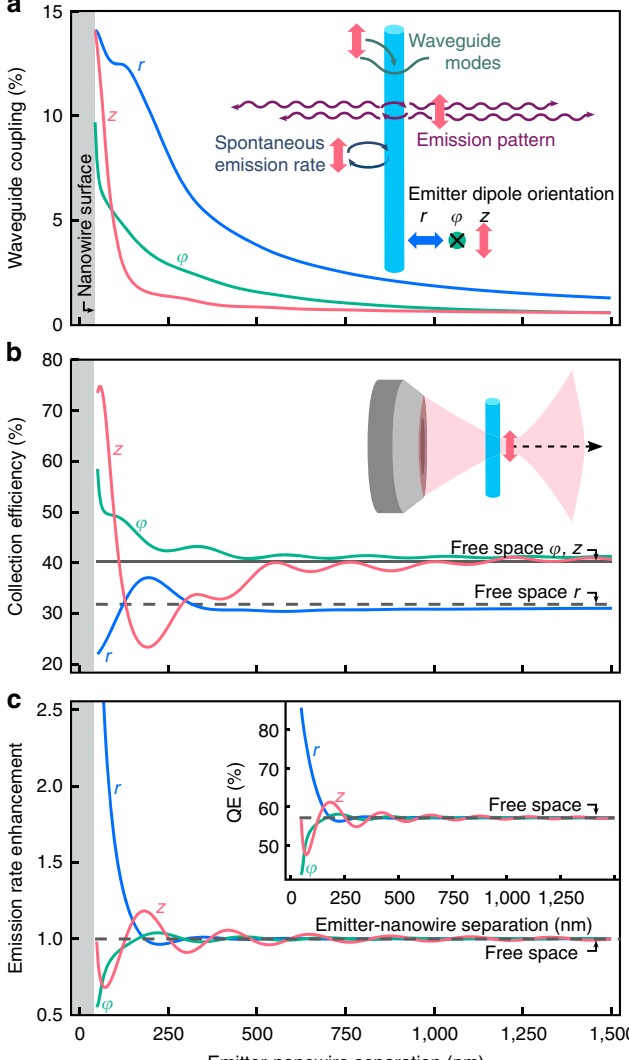

**Figure 1 | Simulations of nanowire–dipole optical interactions.**
(**a**) Fraction of light coupling to nanowire waveguide modes, with colours representing the three cardinal dipole orientations. Inset, schematic summarizing dipole–nanowire optical interactions, of coupling to waveguide modes, interference modifying the emission pattern, emission rate modification, and the cylindrical coordinate dipole orientations.
(**b**) Collection efficiency, measured as the fraction of light emitted in an ∼75° half-angle cone relative to the full sphere, corresponding to the collection of a 0.97 numerical aperture objective lens, compared to free-space dipole emission (solid and dashed black lines). Inset, diagram depicting the simulation. (**c**) Emission rate enhancement relative to the emitter in free-space. Inset, quantum efficiency modification for emitters with a free-space quantum efficiency $\eta_0 = 0.57$. All plots are taken for 648 nm wavelength dipoles with respect to the distance to the centre of a 75 nm diameter silicon nanowire (grey shaded regions), beginning at 5 nm from the surface.

the coupling into the HE$_{11}$ waveguide mode of the wire via the extended evanescent field, even for dipoles 1,500 nm (40 times the radius) away from the surface of the nanowire. This extended evanescent field is also responsible for the exceptionally large absorption cross sections observed in vertically-oriented nanowires, even though those nanowires typically have larger diameters with higher confinement of the guided modes[7,30]. This extended interaction has been essential particularly for the

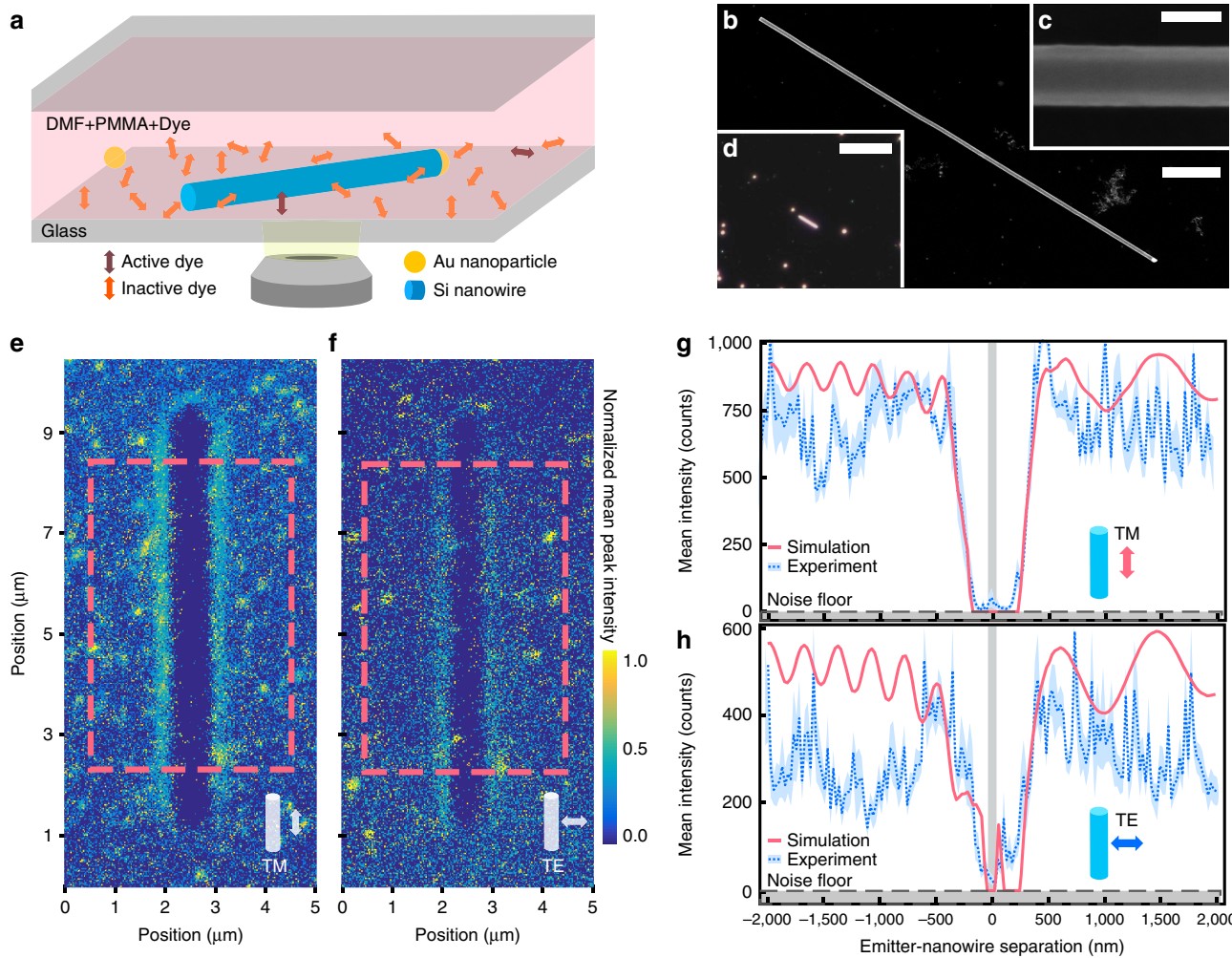

**Figure 2 | Super-resolution-intensity nanowire measurements.** (**a**) Schematic representation of experimental setup, showing nanowire surrounded by fluorophore dye in a solution of poly(methyl methacrylate) (PMMA) and dimethylformamide (DMF), on an inverted microscope. (**b**) Scanning-electron micrograph of the investigated silicon nanowire (scale bar, 1 µm), with (**c**) inset high-resolution image of the smooth wire surface, and ∼ 75 nm diameter (scale bar, 70 nm), and (**d**) dark field optical image of the investigated wire, with surrounding gold nanoparticles visible (scale bar, 20 µm). (**e,f**) 2D super-resolution fluorescence intensity plots of TM and TE fluorescence polarization, respectively, with a 25 × 25 nm² bin averaging. (**g,h**) Corresponding line traces of regions denoted by the dashed red boxes in **e,f** showing the mean intensity in counts of the fluorescence measured until bleached. Simulated traces are overlaid, combining analytical calculations of waveguide coupling, emission rate enactment, directivity modification, point spread function distortion and laser non-uniformity. The vertical grey shaded region depicts the physical nanowire diameter, while the horizontal dashed line corresponds to the noise floor fit in our simulations. The emitter-nanowire separation is measured from the centre of the nanowire. The blue shaded region corresponds to standard error of the mean of the integrated rows.

development of high efficiency photovoltaic and photodetection devices[6–9].

The nanowire also supports localized resonances which strongly scatter[1], leading to interference with the emitter and thus modification of the emission pattern, as visualized in Fig. 1b. Radiation pattern shaping (directivity modification) is of great interest for applications where high yields are required, such as single-photon sources[3,10]. Figure 1b shows the fraction of emitted photons collected by a 0.97 numerical aperture (NA) objective as dipoles move further behind the nanowire (depicted in the inset diagram). We see that contrary to a simple ray-optics view, the collection of emission at small distances from the nanowire is greatly enhanced, reaching 75%, or almost twice the collection efficiency without the nanowire present (dashed and solid black lines).

Finally, Fig. 1c shows the modification to the total emission enhancement of a dipole emitter by the nanowire acting as a nanoscale antenna. This arises from the nanowire modifying the LDOS in the emitter environment, thereby affecting both the quantum efficiency of emitters (inset)[31] and the output rate when emitters are excited at saturation. We show herein that all three of these effects influence measurements of the near field of nanowire optical interactions.

**Experimental imaging of interactions.** The experimental setup used to image these interactions is summarized in Fig. 2a— silicon nanowires are cast onto a glass substrate along with gold nanoparticles used for drift correction and orientation (Fig. 2b–d). The fluorophore solution is then cast onto the substrate and a top glass coverslip is placed to aid focus stabilization and slow solvent evapouration. Caged fluorophores, which are only luminescent after optical activation at higher energies than excitation[32], are used to allow for high total dye

concentrations while maintaining a low active density within a single frame required for super-resolution localization. A polarization filter is inserted in the collection pathway to selectively collect emission from fluorophores oriented either perpendicular (TE) or parallel (TM) to the nanowire axis. PAINT-intensity maps are computed as the average photons per emission event detected, averaged in $25 \times 25\,nm^2$ bins (Supplementary Note 2). In addition, control measurements on silicon oxide nanowires are performed to ensure the measured effects are purely optical and not due to spatial variations in dye concentration from the presence of the structure (Supplementary Note 3).

In the two-dimensional (2D) plots in Fig. 2e,f (showing TM and TE collection polarizations, respectively), we show the obtained PAINT-intensity maps of our nanowire-fluorophore interactions. The large dark regions surrounding the nanowire location make it immediately apparent that the interactions exist at scales far exceeding the nanowire diameter. Bright spots are also visible in these plots, likely due to fluorophore clustering—while we remove the most extreme clustering events during the data processing, the cut-off is somewhat arbitrary and so only events far outside the mean (four standard deviations; Supplementary Note 2) are removed. The influence of this clustering is negligible when averaged between rows and, to better visualize the response, we show line scans integrated over the indicated dashed regions.

The line scans in Fig. 2g,h clarify the large trough of reduced intensity around the wire location, spanning $\sim 13 \times$ the nanowire diameter (shown as the grey vertical shaded region), providing a direct visualization of the nanowire antenna effect. We observe an asymmetric recovery from the extended trough, with the negative positional values expressing a reduced slope relative to the positive positions. In addition, there is a marked difference between the two polarizations, particularly in the existence of a peak within the trough in the TE polarization. While it may be tempting to immediately attribute the presence of the substantial decrease of fluorescence intensity to excitation of waveguide and resonant modes in the nanowire, the multitude of possible phenomena discussed in Fig. 1 suggests that the true interactions are more complex.

**Modelling of interactions**. In addition to the three main interactions discussed in Fig. 1, we incorporate a calculation of the dependence of the field from the laser excitation on the location relative to the nanowire—this occurs from interference of the excitation laser with the field scattered by the nanowire. In addition, the laser enters the experimental setup at an $\sim 30°$ angle to limit back reflections, concomitantly producing an asymmetric response on the left and right sides of the wire. Furthermore, the directivity modifications shown in Fig. 1b influence our experimental measurements not only as a modification of the observable peak brightness, but also as a distortion of the point spread function (PSF), shifting the peak location from the true fluorophore position away from the wire when close to the wire surface. These effects are also calculated and included in our simulations (Supplementary Note 4).

By first comparing our simulated and experimentally measured trends in detected brightness as a function of the emitter position relative to the nanowire centre, as shown in Fig. 2g,h, we can establish the validity and completeness of our theoretical models. In Fig. 2g,h, we incorporate a single fitting parameter between our model and experimental data, equivalent to a noise floor—this is necessary as sufficiently dim fluorescence events calculated by our model would not be possible to localize in an experiment. After the determination of the centre position and

noise floor fit between the TM polarization model and experimental data, these parameters are used (not re-fit) for running the simulation of the TE polarization. This ensures that the only difference between the two simulations is indeed the polarization being simulated. Doing so, we observe good agreement between main features of the dim regions—the width, asymmetric steepness of the trough sides, and the periods of the oscillations on negative-valued side of the wire (327 and 359 nm experimental for TM and TE, respectively, versus 320 nm in the simulation, as determined through Fourier analysis; Supplementary Fig. 3a). We additionally recover the observed peak within the trough in this polarization, and observe evidence of the flat region at $\sim -200\,nm$ in the TE polarization, further supporting the validity of the simulations to capture the rich interactions present in the experimental traces. The confirmation of such features also demonstrates the necessity of super-resolution measurements—the absence of such features (as in a diffraction-limited measurement, Supplementary Note 5) would preclude any assurance of the agreement with theory.

**Deconvolution of interactions**. The observed agreement between the simulations and experiments furthermore allows for the extraction of parameters not directly measurable experimentally (for example, waveguide mode excitation) with a reasonable level of confidence. To this end we compute the power distribution into the various channels (emission modification, waveguide mode excitation and collected emission) present in our measurements.

In Fig. 3, we show the relative contribution of the modelled optical phenomena to our PAINT-intensity traces. In these calculations we begin with the total emission from a dipole as modified by the proximity to the nanowire (top blue trace). We then remove the power coupled into waveguide modes (blue shaded region) followed by the power not collected due to redirection (pink shaded region), resulting in the red trace. The influence of the image distortion causing displacement of the PSF from the true dipole location is finally included in the bottom green trace. For clarity we discount the influence of the laser excitation non-uniformity in the analysis here.

We first note that the emission of the dipole due to the LDOS enhancement from the wire (modifying the quantum efficiency and rate of emission) is substantially different between the two polarizations—the TM polarization shows an oscillating behaviour from the vertically-oriented ($z$) dipoles expressing a Drexhage-like effect in emission[13], as is also observed in Fig. 1c. The TE polarization, however, shows a strong enhancement of emission, largely due to the substantial LDOS enhancement of radially-oriented ($r$) dipoles near the wire.

While it is observable that waveguide coupling of the dipole emission (displayed as the dark blue shaded region in Fig. 3a,b) is not the dominant influence in the observed interactions, the contribution is still important; the TM polarized collected emission (corresponding to $z$-oriented dipoles in Fig. 1a) shows strong coupling immediately adjacent to the wire, with the contribution falling off rapidly beyond $\sim 100\,nm$. Conversely, for the TE polarized collected emission (largely $r$-oriented dipoles in Fig. 1a), the waveguide coupling is notable even beyond 1,000 nm. The excitation of Mie resonances (responsible for both broadening and distorting/displacing the PSF of the image) contributes more notably to the reduction in the peak intensity (pink shaded area in Fig. 3a,b).

Additionally, while large displacements influence the green trace in the TM polarization quite significantly, they play almost no role in the final trace of the TE polarization, indicating that polarization control can be useful in limiting the observation

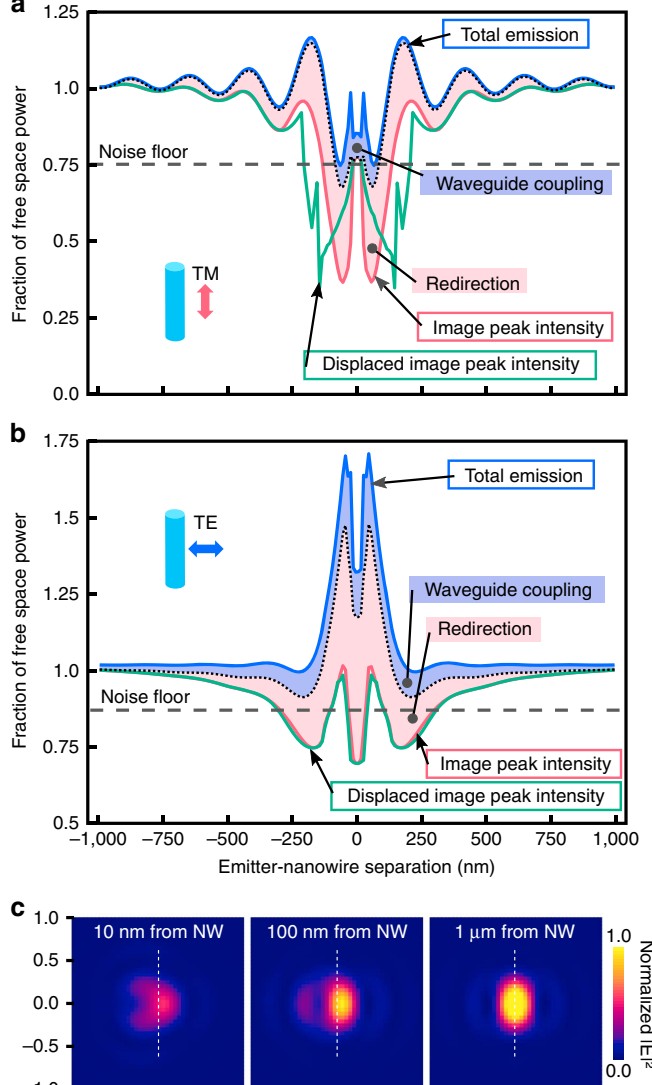

**Figure 3 | Deconvolved power distributions.** Simulated power distributions for TM, (**a**) and TE, (**b**) polarizations. Neglecting the excitation laser influence for clarity, fluorophore total emission due to enhancement (or quenching) by the wire (blue line) and the observable peak intensity of a Gaussian fit to the point spread function (PSF) image (pink line) are shown. The power difference between these two traces is shown divided between the coupling to waveguide modes and the redirection of emission by scattering (blue and pink shaded regions, respectively). The above are taken with respect to the true dipole positions, with the final bottom green trace showing the locations when the PSF displacements are included. (**c**) Image plane simulations of PSF modification by interaction of a z dipole with the nanowire, showing peak displacement from the true dipole position (white line) as well as geometric distortions (which also decrease the peak intensity).

of PSF distortions (depicted in Fig. 3c) in such measurements where this influence is not of interest. Recent work has also shown metasurfaces to have promising applications for the limitation of PSF shape influences[33]. Alternatively, techniques could be employed to simultaneously measure the position and orientation of the imaged dipoles as demonstrated previously[34], thereby allowing additional control over the imaged field components and orientations.

## Discussion

These measurements provide new experimental evidence of the nature of the long range interactions present between dipole-like emitters and dielectric nanostructures, and offer a novel characterization method to quantify these interactions at high-resolution, without the need for external near-field probes.

Here, we measure the ensemble of interactions between fluorophores and nanostructures. Further improvements to the method, particularly those discussed in limiting PSF distortions as well as combinations with lifetime measurements[13,24], could be implemented. Combined, these methods could permit detailed mapping of individual optical components, providing a valuable new tool in understanding the near-field environments of nanophotonic objects.

## Methods

**Sample preparation.** The silicon nanowires are grown epitaxially on a silicon substrate by a vapour-liquid-solid growth mechanism using an atmospheric-pressure chemical vapour deposition system. 100 nm gold colloids (BBI Solutions) were used as catalysts. The growth was conducted at 850 °C for 20 min with silicon tetrachloride ($SiCl_4$) as the precursor. The carrier-gas flows during growth are Ar = 95 s.c.c.m. and $H_2$ = 30 s.c.c.m., while 19 s.c.c.m. Ar gas flows directly through the silicon tetrachloride precursor bubbler (held at 0 °C in a temperature-controlled bath).

The clean coverslips are prepared by soaking overnight in Hellmanex detergent, sonicating for 15 min in each acetone and ethanol, and soaking in 50% hydrogen peroxide overnight, with rinsing in Milli-Q purified water after each step, and finally drying in nitrogen. Wires are exfoliated in isopropanol and cast on the clean glass coverslips. Fiducial marks are scribed into the sample for rough wire position determination, and additional gold nanoparticles are cast as well for drift correction and to aid in fine positional wire location between super-resolution and SEM measurements.

**Fluorophore solution preparation.** Solutions of 0.01 g l$^{-1}$ CAGE 635 fluorophore (Abberior) in dimethylformamide (DMF) with a 5 wt% poly(methyl methacrylate) (PMMA) 395,000 molecular weight polymer (Sigma-Aldrich) are mixed immediately prior to imaging, to limit dye clustering. PMMA is used to increase solution viscosity, limit dye desorption and (rotational) diffusion, and ensure collection during full dye bleaching. The solution dye concentration is calibrated to remain low enough to ensure only single localizations within a diffraction limited spot, but high enough to allow for dense sampling over the area of interest within a reasonable timeframe (∼30 min measurements) before clustering or solvent evapouration become overly problematic.

Approximately 25 μl of the dye solution is sandwiched between two clean glass coverslips to limit evapouration of the DMF solvent and improve stability of the instrument's active focus stabilization system. The dye forms a layer ∼1 μm in thickness as measured by the calibrated focus system in the microscope. This is far larger than the depth of field of the employed objective and thus has no influence on the localization.

**Super-resolution measurements.** The measurement setup consists of a Nikon Ti Eclipse inverted microscope with an oil (n = 1.515) immersion objective of NA = 1.49 (although the usable NA is likely closer to 1.46) (ref. 35), and an electron-multiplying charge-coupled device (EMCCD) camera (Andor iXon Ultra 897). A custom-made polarization filter can be optionally inserted to limit collection polarization to a particular orientation with respect to the wire.

Dye molecules are activated by a 405 nm wavelength activation laser, at an intensity of ∼40 W cm$^{-2}$, and pumped until bleached with a 640 nm excitation laser, at an intensity of ∼400 W cm$^{-2}$. Laser intensities are chosen to allow a small number of dye molecules (much fewer than 1 per square micron) to be activated in any given frame, while pumping most active dye molecules until bleached within the single frame integration time of 95 ms. Emissions lasting multiple frames are merged to measure total brightness. Band pass filters are placed in the collection path to limit detection of the activation and excitation lasers, while still collecting the 648 nm peak activated dye emission.

The low localization uncertainty (50 nm$^2$ mean squared error) yet high mean-square displacement when free in solution (10 μm$^2$ in a 95 ms frame), implies that events are localized only when stochastically adsorbed onto a surface (Supplementary Note 6).

Measurements consist of ∼30,000 frames, corresponding to ∼300,000 localization events within an ∼15 × 15 μm$^2$ area of interest around a nanowire. After collection of the image frames, the PSF localization is performed using the ThunderSTORM v1.3 package[36]. After all optical measurements are complete (to prevent carbon contamination), samples are imaged in a scanning-electron microscope to determine the precise geometry of the interrogated structure. These measurements are used in the analytical modelling of the system.

For additional details on the data processing and analysis see Supplementary Note 2.

**Analytical model.** In order to determine the contributions of the optical effects in our system we create a semi-empirical Monte-Carlo simulation of imaging conditions: A 1D array of 300,000 virtual fluorophores is established, with their position uniformly distributed along the surface of the substrate and nanowire (in agreement with the larger number of localizations around the wire observed in control measurements; see following section). For each fluorophore a random dipole orientation is assigned.

Analytical Green's function and Lorenz-Mie theory calculations are used to deconvolve the possible interactions. Specifically, we incorporate the excitation power of the laser (and its interference with the nanowire), the quantum efficiency and LDOS enhancement of the fluorophore by the nanowire, the absorption in the wire, the scattering/collection modification, and the distortion of the PSF allowing possible displacements in the imaged versus true location of the fluorophore. All calculations are implemented in the Julia language[37].

For additional details on the simulations and calculation methods, see Supplementary Note 4.

**Data availability.** The data that support the findings of this study are available from the corresponding author on reasonable request.

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

## Acknowledgements

We would like to thank M. Kamp and A. F. Koenderink (AMOLF) for helpful discussions. This work is part of the research program of the Foundation for Fundamental Research on Matter (FOM), which is part of The Netherlands Organization for Scientific Research (NWO). We acknowledge financial support from the European Research Council under the European Union's Seventh Framework Programme (FP/2007-2013)/ERC grant agreement no. 337328, 'NanoEnabledPV' and by a TKI instrumentation grant together with FEI.

## Author contributions

E.C.G., E.J., J.S. and T.S.S. designed the experiments. E.J. and J.S. carried out the experiments. J.W. provided the nanowire materials. E.J. and S.A.M. designed and carried out the simulations. E.J. processed, analysed and interpreted the data. All authors contributed to the writing of the manuscript.
