## [Peer Review File · Nature Communications]

Reviewers' comments:

Reviewer #1 (Remarks to the Author):

In this paper the authors report on a novel method to contactless image emitter-nanostructure interactions, demonstrated using Si nanowires as test objects. The experiments are sound and the results are interesting, revealing near field optical interactions and coupling between dipole-like sources and nanowire antennas. Referencing is appropriate, and the text is clear.

I do have some concerns which when considered merits publication in my opinion.

I do not understand the argumentation line 89-90, using the word lossless.

Line 96-97. I do not see how the near field can "explain" the large absorption cross section, indicate, show etc are more accurate in my opinion.

Fig 1: for the reader, specify that this is modelled results solely, also in the legend. To me, it was not immediately clear where the nanowire was, nor how the different lines with different colors should be interpreted. In figures, do not use color as the only way of identifying which line is which, think about the color blind.

Line 109, should emitters be emission?

Fig 2, to me its not clear why some of the dye is indicated inactive and other active.

Fig 2, the catalytic Au particle seems to be left on the NW in fig 2b, not correlating with the schematic in fig 2a

Fig 2 g, simulation in the vicinity of 0 distance deviates significantly from the experiment. Please comment.

Fig 2, h, the simulation shows a similar trend as the experiment, but the values for distances beyond 500 nm deviate between experiment and simulation up to almost 200%?

Fig 3, specify that these are modelled data solely in the legend.

Since the paper includes a large part of modelled data, please include a short description of how the simulations were carried out also in the main part of the paper, not only in the SI.

line 70, consider skipping word "new" before super resolution technique, its obvious.

line 125 aide should be aid

line 164, "suggest the" should probably be "suggest that the"

Fig S1, spelling error within graph "limted".

Fig S2, don't use color only for differentiating between lines.

Fig S3, why not simply cut out the imaged part in a) and rotate it such that it correlates to the data in b and c.

Reviewer #2 (Remarks to the Author):

This is a clearly written article that reports a creative approach to imaging the near-field coupling between single molecules and a silicon nanowire, namely, immersing that nanowire in a solution of dye molecules and imaging the field of dye using PALM. The argument is clearly and logically presented and the comparison between simulations and experimental data is excellent and displayed

compellingly in Figure 2. The experimental observation of the peak within the trough and its agreement with the simulations is especially compelling.

I recommend publication after minor improvement to the figures and presentation of experimental methods. I do not need to see the manuscript again; it is fine for the editors to make sure that these improvements have taken place.

Recommended revisions to figures:

All figures would be improved by more careful labeling that allows a reader who prints a black and white copy can tell which trace is which. Presently it's not possible to interpret the figures if printed in black and white. For example, in Figure 1, the letters R, phi, and Z can be placed next to the corresponding traces (perhaps also putting those labeling letters in the corresponding colors). It might be a little tricky to lay this out in panel c, but it would be very straightforward in panels a and b, and it will help all readers parse the figure, even those who have it in color.

In Figure 1b, the schematic of the directional emission is not mentioned anywhere either in the text or the figure caption. This should be described somewhere.

In Figure 2g and h, the green dots indicating the width of the nanowire are hard to see and understand. I suggest instead demarcating the nanowire with a shaded area as is provided at the left of Figure 1.

Recommended revisions to methods (in supplementary materials):

I have not been able to find the volume of fluorophore solution that is used in sample preparation or the approximate thickness of the resulting volume once it is covered with a cover slip. The authors should add the volume used and a discussion of how they know that their results are not affected by the presence of the top cover slip (the one used to avoid evaporation, not the one on which the nanowire rests).

In section A of the supplementary materials, the authors refer to the nanowires as being cast on glass "slides". However, these must be cover slips as the microscope being used is an inverted microscope, and later the authors refer to the sample solution as being sandwiched between two glass cover slips. This should be corrected. In addition, the authors should describe how they cleaned the cover slips (or if they used them straight out of the box – they refer to "clean glass slides" or "clean cover slips").

Reviewer #3 (Remarks to the Author):

In this work Johlin et.al. introduce microscopy-based technique to image emitter-nanostructure interactions. For a 75 nm diameter silicon nanowire, authors observe a confluence of emission rate enhancement, directivity modification, and guided mode excitation, with strong interaction at scales up to 13 times of the nanowire diameter. Authors also provide a thorough analytical modeling that allows them to distinguish the relative contribution of these effects, as well as their dependence on emitter orientation. These effects are mostly due to the long range interactions between the dielectric surface of the NW and dipole-like emitters. These effects are systematically studied for both polarizations TM and TE.

The manuscript is well structured and should be of interest to the community working with nanostructures. However, several sections of the paper should be improved and clarified:

In localization microscopy dipole anisotropy of the imaging probes has been studied before. Authors have neglected to cite and to study previous work that also exploits simultaneous imaging of the positions and fluorescence anisotropies of large numbers of single molecules with nanometer lateral resolution within a sample (see Nature Methods. 2008. Gould et.al).

The authors should cite more of the localization microscopy literature than just the work from Hess et al (ref 28).

(the proper citation for any single molecule localization microscopy are three seminal works from 2006.

1. Betzig, E. et al. *Science* 313, 1642–1645 (2006).
2. Rust, M.J., Bates, M. & Zhuang, X. *Nat. Methods* 3, 793–795 (2006)
3. Hess, S.T., Girirajan, T.P. & Mason, M.D. *Biophys. J.* 91, 4258–4272 (2006).

- It is mentioned that the emitter-nanostructure interactions are mapped with a resolution around 25 nm. How was the resolution determined? It should not be confused with the localization error.
- In relation to this comment, what is meant with 50 nm standard error on page 2? Is it the localization error? Also, it is not clear to me what the 10 μm^2 mean square displacement in a 95 ms frame means. Are the authors referring to tracking of the molecules? How can this be done within a single frame?
- On page 2 it is stated that “events are localized only when stochastically adsorbed onto a surface”. This type of localization microscopy strongly resembles the PAINT approach, and not the classical PALM concept. The authors should make this clearer and probably refer to their technique in a different way than “PALM”.
- How exactly are the super-resolution intensity images in figure 2e-f generated? By plotting the average number of emitted photons per emitter in a 25 nm bin? This should be clarified.
- What is the reason for the bright “spots” on the more uniform dimmer background in figure 2e-f? Couldn't they be causing the oscillations in the line averages in figure 2g-h? This could be excluded by imaging more nanowires
- In figure 3. Authors present Image plane simulations of PSF modification by interaction of a Z-dipole with the A nanowire, showing peak displacement from the true dipole position (white line) as well as geometric distortions (which also decrease the peak intensity. It would be beneficial to present as well experimental PSFs for comparison

Point-by-point responses to reviewers' comments

(Responses inset and in red)

Reviewer #1 (Remarks to the Author):

In this paper the authors report on a novel method to contactless image emitter-nanostructure interactions, demonstrated using Si nanowires as test objects. The experiments are sound and the results are interesting, revealing near field optical interactions and coupling between dipole-like sources and nanowire antennas. Referencing is appropriate, and the text is clear.

I do have some concerns which when considered merits publication in my opinion.

I do not understand the argumentation line 89-90, using the word lossless.

For 648 nm radiation, the mode is largely contained outside of the 75 nm diameter wire. The complex part of the refractive index is also quite low, thus there will be little absorption or “loss” in the wire, leading us to explain that the wire is essentially “lossless.” For larger wires or shorter wavelengths (higher confinement leading to higher modal overlap with the wire physical area, and higher complex refractive index) this would not be the case, which we mention as the conditions that lead to strong absorption in lines 95-99 of the original manuscript.

The phrasing however was indeed confusing, and we thus have reworded this passage to: “For this size and wavelength absorption in the nanowire is minimal and can be neglected (see Supplementary Note 1), such that we only have to consider scattering and excitation of waveguide modes.”

Line 96-97. I do not see how the near field can “explain” the large absorption cross section, indicate, show etc are more accurate in my opinion.

A good point – this has been modified to read “This extended evanescent field is also responsible for the exceptionally large absorption cross sections [...]” accordingly.

Fig 1: for the reader, specify that this is modelled results solely, also in the legend. To me, it was not immediately clear where the nanowire was, nor how the different lines with different colors should be interpreted. In figures, do not use color as the only way of identifying which line is which, think about the color blind.

We have adjusted the figures and legends accordingly.

Line 109, should emitters be emission?

Indeed, this has been corrected.

Fig 2, to me its not clear why some of the dye is indicated inactive and other active.

We use CAGE dye, where essentially all dye begins in an inactive state, and two wavelengths are used, first to bring a small portion of the dye from inactive to active, and a second to bring only the active dye to a fluorescent state. This allows high density of dye to be used while still only sampling a very small fraction of dye, and thereby still satisfying the requirement of only a single fluorescence even within a diffraction limited spot in any given frame.

This is explained in lines 126-131, but described as “caging” and “uncaging” in the text, which we agree is confusing with the diagram. We have re-written this section to be more clear, referring to the uncaging process as “activation” now.

Fig 2, the catalytic Au particle seems to be left on the NW in fig 2b, not correlating with the schematic in fig 2a

The catalytic Au particle was originally neglected as we exclude this region from most of our analysis, but we agree the difference between the image and diagram could be confusing and so we have added this to the diagram as well.

Fig 2 g, simulation in the vicinity of 0 distance deviates significantly from the experiment. Please comment.

The deviation in Fig. 2 **g** is likely due to the noise floor (indicated by the dashed line). In a real measurement the values would simply be zero for the area below this line (and not become negative as shown in the previous version of the figure). We agree this appears confusing as to why the simulation becomes negative (as this is essentially unphysical). To remedy this we have simply confined the simulation to positive values, causing the agreement to become more obvious.

Fig 2, h, the simulation shows a similar trend as the experiment, but the values for distances beyond 500 nm deviate between experiment and simulation up to almost 200%?

Fig. 2 **h** contains somewhat fewer data points than Fig. 2 **g** (which can be seen to have much better agreement in the background region), due to the measurement being performed after and thus having fewer unbleached fluorophores available. The averaging and analysis was however kept constant between the two panels, particularly to allow better comparison around the nanowire region. Essentially this deviation is due to the larger presence of “zero” value pixels in panel **h**.

We should note that if further averaging is done, as the zeros are removed in the background area, these values converge to ~520 counts, in agreement with the simulations. However, details of the trough are lost at this level of averaging. Because the far regions past ~600 nm have little difference in terms of interactions with the nanowire between the TM and TE polarizations (**g** and **h**), we believed that keeping the analysis the same, and showing the variation in the trough region was the best choice for the analysis.

A discussion addressing this point has been added to the Supplementary Information as well.

Fig 3, specify that these are modelled data solely in the legend.

Since the paper includes a large part of modelled data, please include a short description of how the simulations were carried out also in the main part of the paper, not only in the SI.

We have adjusted the legends accordingly.

The new inclusion of a Methods section (with a subsection on the analytical simulations) in the main paper largely addresses the request to put more of the modeling information in the main text. Additionally, we have included a sentence in the first paragraph of the results section, mentioning “These interactions can be calculated using commercial software packages (FEM, FDTD) or, as in this case, solved analytically using a Green's function.”

line 70, consider skipping word “new” before super resolution technique, its obvious.

line 125 aide should be aid

line 164, “suggest the” should probably be “suggest that the”

Fig S1, spelling error within graph “limted”.

These four comments have been incorporated. We thank the reviewer for their careful reading in spotting these errors.

Fig S2, don't use color only for differentiating between lines.

Fig S3, why not simply cut out the imaged part in a) and rotate it such that it correlates to the data in b and c.

These suggestions for figure modifications have also been incorporated.

Reviewer #2 (Remarks to the Author):

This is a clearly written article that reports a creative approach to imaging the near-field coupling between single molecules and a silicon nanowire, namely, immersing that nanowire in a solution of dye molecules and imaging the field of dye using PALM. The argument is clearly and logically presented and the comparison between simulations and experimental data is excellent and displayed compellingly in Figure 2. The experimental observation of the peak within the trough and its agreement with the simulations is especially compelling.

I recommend publication after minor improvement to the figures and presentation of experimental methods. I do not need to see the manuscript again; it is fine for the editors to make sure that these improvements have taken place.

Recommended revisions to figures:

All figures would be improved by more careful labeling that allows a reader who prints a black and white copy can tell which trace is which. Presently it's not possible to interpret the figures if printed in black and white. For example, in Figure 1, the letters R, phi, and Z can be placed next to the corresponding traces (perhaps also putting those labeling letters in the corresponding colors). It might be a little tricky to lay this out in panel c, but it would be very straightforward in panels a and b, and it will help all readers parse the figure, even those who have it in color.

This is a great suggestion, and addresses concerns of other reviewers as well – we have accordingly modified Figure 1 and (the originally numbered) Figure S2 in this way to make the identity of the lines more clear.

In Figure 1b, the schematic of the directional emission is not mentioned anywhere either in the text or the figure caption. This should be described somewhere.

We have added a note to the legend of Fig. 1 mentioning the inset in 1b is a diagram depicting the simulation carried out in that panel, as well as a note in the text as well.

In Figure 2g and h, the green dots indicating the width of the nanowire are hard to see and understand. I suggest instead demarcating the nanowire with a shaded area as is provided at the left of Figure 1.

This is a good point, and we have modified the figure to include the shaded region instead of the green dot as suggested to maintain similarity between Figs. 1 & 2.

Recommended revisions to methods (in supplementary materials):

I have not been able to find the volume of fluorophore solution that is used in sample preparation or the approximate thickness of the resulting volume once it is covered with a cover slip. The authors should add

the volume used and a discussion of how they know that their results are not affected by the presence of the top cover slip (the one used to avoid evaporation, not the one on which the nanowire rests).

This was an oversight on our part, and has now been included – the volume of solution was 25 μL , and the thickness of the layer is approximately 1 μm . This was measured by moving the focal plane from the bottom surface (the nanowire and adsorbed dye molecules are clearly visible) up to the top slide surface (dye adsorption becomes visible again, while the nanowire is no longer in focus). The focal plane in the microscope is calibrated, allowing us to measure the gap between the two coverslips filled with the solution.

The depth of field of the microscope system is also much narrower than this gap (~150 nm), and includes a feedback system to maintain focus on the lower interface, suggesting the top coverslip will not influence the measurement.

This discussion has been added to the Methods section.

In section A of the supplementary materials, the authors refer to the nanowires as being cast on glass “slides”. However, these must be cover slips as the microscope being used is an inverted microscope, and later the authors refer to the sample solution as being sandwiched between two glass cover slips. This should be corrected. In addition, the authors should describe how they cleaned the cover slips (or if they used them straight out of the box – they refer to “clean glass slides” or “clean cover slips”).

This was indeed a mistake. We have adjusted all mentions of the substrate to “coverslips.”

The clean coverslips are prepared by soaking overnight in Hellmanex detergent, sonicating for 15 min in each acetone and ethanol, and soaking in 50% hydrogen peroxide overnight, with rinsing in Milli-Q purified water after each step, and finally drying in nitrogen.

The cleaning procedure has also been added to the Methods section.

Reviewer #3 (Remarks to the Author):

In this work Johlin *et.al.* introduce microscopy-based technique to image emitter-nanostructure interactions. For a 75 nm diameter silicon nanowire, authors observe a confluence of emission rate enhancement, directivity modification, and guided mode excitation, with strong interaction at scales up to 13 times of the nanowire diameter. Authors also provide a through analytical modeling that allows them to distinguish the relative contribution of these effects, as well as their dependence on emitter orientation. These effects are mostly due to the long range interactions between the dielectric surface of the NW and dipole-like emitters. These effects are systematically studied for both polarizations TM and TE. The manuscript is well structured and should be of interest to the community working with nanostructures. However, several sections of the paper should be improved and clarified:

In localization microscopy dipole anisotropy of the imaging probes has been studied before. Authors have neglected to cite and to study previous work that also exploits simultaneous imaging of the positions and fluorescence anisotropies of large numbers of single molecules with nanometer lateral resolution within a sample (see Nature Methods. 2008. Gould *et.al.*).

The authors should cite more of the localization microscopy literature than just the work from Hess *et al* (ref 28).

(the proper citation for any single molecule localization microscopy are three seminal works from 2006.

1. Betzig, E. *et al.* Science 313, 1642–1645 (2006).
2. Rust, M.J., Bates, M. & Zhuang, X. Nat. Methods 3, 793–795 (2006)
3. Hess, S.T., Girirajan, T.P. & Mason, M.D. Biophys. J. 91, 4258–4272 (2006.)

We thank the reviewer for their careful review of our references and knowledge of this field, and have accordingly included these works in our references.

- It is mentioned that the emitter-nanostructure interactions are mapped with a resolution around 25 nm. How was the resolution determined? It should not be confused with the localization error.

The resolution mentioned here is simply the pixel size used to determine the plots in Fig. 2 e-h.

It indeed should be noted that with a mean squared error in the localization precision of a fluorescence event of 50 nm^2 , this can be thought of as a Gaussian function (probability distribution) with a standard deviation of $\sim 7 \text{ nm}$ (and full width at half max [FWHM] of $\sim 17 \text{ nm}$). This is well within the pixel size we use, so the resolution of our maps is simply limited by our pixel size.

- In relation to this comment, what is meant with 50 nm standard error on page 2? Is it the localization error? Also, it is not clear to me what the $10 \mu\text{m}^2$ mean square displacement in a 95 ms frame means. Are the authors referring to tracking of the molecules? How can this be done within a single frame?

This was indeed referring to the localization uncertainty, or mean squared error, and has been referred to as such now to avoid confusion. The units should be nm^2 however, and this has been corrected.

The “mean square displacement” discussion was in reference to the average distance a fluorophores molecule will diffuse when free in the solution (if not adsorbed) and is simply calculated to support the view that the measurements are occurring on fluorophores adsorbed on a surface. This has been clarified in the text, as well as more clearly referencing the Supplementary Note 6 where this calculation is discussed.

These have also both been moved to the Methods section to improve the flow of the manuscript, and avoid possible confusion.

- On page 2 it is stated that “events are localized only when stochastically adsorbed onto a surface”. This type of localization microscopy strongly resembles the PAINT approach, and not the classical PALM concept. The authors should make this clearer and probably refer to their technique in a different way than “PALM”.

This is an extremely useful insight, and we have included a reference to the PAINT approach in the last paragraph of the introduction, as well as modifying the text to replace the denotation of our measurement as “PALM” to “PAINT,” and our mapping as “PAINT-intensity” instead of “PALM-intensity,” as suggested.

- How exactly are the super-resolution intensity images in figure 2e-f generated? By plotting the average number of emitted photons per emitter in a 25 nm bin? This should be clarified.

The reviewer is correct, these are averaged photon counts in a 25 nm bin. This was added to the text, stating that “PAINT-intensity maps are computed as the average photons per emission event detected, averaged in 25 nm bins (see Supplementary Note 2).”

- What is the reason for the bright “spots” on the more uniform dimmer background in figure 2e-f? Couldn't they be causing the oscillations in the line averages in figure 2g-h? This could be excluded by imaging more nanowires

The bright spots appear to be dye clusters present in the solution. While these could be removed by sectioning of the experimental data, the selection criteria for what is or is not a cluster would be somewhat arbitrary, and so we err on the side of including data instead of removing any possible clustering.

The observation of the oscillations in Fig.2 **g-h** are unlikely to be due to the bright spots, as the same period is observed in both map (see Supplementary Fig. 3a insets). Additionally, if we partition the data into two equal regions (top and bottom) we still observe the same frequency-domain response (a dominant peak around 350 nm).

- In figure 3. Authors present Image plane simulations of PSF modification by interaction of a Z-dipole with the A nanowire, showing peak displacement from the true dipole position (white line) as well as geometric distortions (which also decrease the peak intensity). It would be beneficial to present as well experimental PSFs for comparison

This is a good point – while images of the distortions present in Fig. 3c would be largely uninteresting (they are essentially displacements of the point spread function), we have included a new figure in the Supplementary Information (Supplementary Figure 4) showing a more clear PSF distortion. These, while not contributing to the displacement influences discussed in the text, establish the sensitivity of our measurement setup to such distortions, as well as the similarity between the modeled distortions and their observable occurrence in our measurements.

REVIEWERS' COMMENTS:

Reviewer #1 (Remarks to the Author):

My concerns have been met and I recommend publication.

Reviewer #3 (Remarks to the Author):

My comments have been fully addressed

REVIEWERS' COMMENTS:

Reviewer #1 (Remarks to the Author):

My concerns have been met and I recommend publication.

Reviewer #3 (Remarks to the Author):

My comments have been fully addressed

AUTHORS' RESPONSE:

We are very pleased to see that all previous comments and concerns have been fully addressed.

All included changes are in response to editorial requests.